# Monitoring of *Fusarium* Species and Trichothecene Genotypes Associated with *Fusarium* Head Blight on Wheat in Hebei Province, China

**DOI:** 10.3390/toxins11050243

**Published:** 2019-04-28

**Authors:** Lijing Ji, Qiusheng Li, Yajiao Wang, Lester W Burgess, Mengwei Sun, Keqiang Cao, Lingxiao Kong

**Affiliations:** 1Plant Protection Institute, Hebei Academy of Agricultural and Forestry Sciences, Key Laboratory of Integrated Pest Management on Crops in Northern Region of North China, Ministry of Agriculture, Baoding 071000, China; jilijing79@hotmail.com (L.J.); qiushengli1978@hotmail.com (Q.L.); yajiaowang0515@outlook.com (Y.W.); sunmengwei1994@outlook.com (M.S.); 2School of Biological Sciences, Faculty of Science, University of Sydney, Sydney 2006, New South Wales, Australia; burgess.international@gmail.com; 3College of Plant Protection, Agricultural University of Hebei, Baoding 071001, China

**Keywords:** Fusarium head blight, trichothecene genotype, pathogen composition, monitoring

## Abstract

To clarify the changes in field populations of Fusarium head blight (FHB) pathogens over a decade, *Fusarium* species and trichothecene genotypes associated with FHB on wheat were monitored in Hebei province during the periods 2005–2006 and 2013–2016. *Fusarium* species determination was carried out by morphological identification, species-specific amplification and partial translation elongation factor (TEF-1α) gene sequencing. Trichothecene genotype prediction was carried out by primers 3CON/3NA/3D15A/3D3 or Tri13F/Tri13R, Tri303F/Tri303R and Tri315F/Tri315R. A total of 778 purified *Fusarium* isolates were recovered from 42 sampling sites in 17 counties during the period 2005–2006 and 1002 *Fusarium* isolates were recovered from 122 sampling sites in 65 counties during the period 2013–2016. *F. graminearum* was the predominant pathogen recovered during the periods 2005–2006 and 2013–2016. However, the pathogen composition differed slightly between the two periods. In 2005–2006, 752 out of 778 (96.7%) of the isolates belonged to *F. graminearum*. Two were identified as *F. culmorum*. Five other *Fusarium* species were also recovered, *F. equiseti*, *F. verticillioides*, *F. proliferatum*, *F. subglutinans* and *F. chlamydosporum*, with lower recoveries of 0.4%, 0.8%, 0.8%, 0.1% and 1.0%, respectively. Trichothecene genotype prediction showed that all the 752 *F. graminearum* isolates were of the 15-ADON genotype. Five *Fusarium* species were recovered from samples collected over the period 2013–2016. *F. graminearum* was again the predominant pathogen with an isolation frequency of 97.6%. *F. pseudograminearum*, *F. asiaticum*, *F. culmorum* and *F. negundis* were also isolated at a recovery of 1.4%, 0.7%, 0.2% and 0.1%, respectively. For the 2013–2016 isolates, 971 of the 978 *F. graminearum* strains were 15-ADON whereas seven isolates were of the 3-ADON type. All seven *F. asiaticum* isolates were of the NIV type and fourteen *F. pseudograminearum* isolates were classified as 3-ADON. *F. pseudograminearum* was first isolated from FHB in Hebei in 2013. Although the recovery of *F. pseudograminearum* is still low, it represents a small shift in the pathogen composition and trichothecene genotypes associated with FHB in Hebei province. As Fusarium crown rot of wheat caused by *F. pseudograminearum* is an increasing problem in Hebei province, it is appropriate to monitor the role of *F. pseudograminearum* in FHB in the future.

## 1. Introduction

Fusarium head blight (FHB), caused by a complex of the *Fusarium* species, is a devastating disease affecting wheat and other small grains in many regions throughout the world [1]. In addition to direct yield losses, FHB reduces grain quality, and harvested grain is often contaminated with mycotoxins [2]. This disease affects wheat, barley and other small grains both in temperate and subtropical regions. It has the capacity to destroy a potentially high-yielding crop within a few weeks of harvest [1]. During the early years of the twentieth century, FHB was considered a major threat to wheat and barley worldwide. In China, FHB caused epidemics in the mid- to downstream regions of the Yangtze River, the northeast province of Heilongjiang, and parts of southern China in the 1950s to 1960s [3].

In recent decades, FHB has again become a major problem worldwide and epidemics have occurred in Asia, Australia, Europe, southern Africa and North America [4,5,6]. In the mid 1990s, wheat losses from FHB were estimated at $1–2.5 billion in the USA, $220–300 million in Canada and caused major socio-economic problems [1,7]. Similarly, in 1985, a major epidemic of FHB was also reported in Henan province on the North China Plain (the main wheat-growing region of China), affecting 3.7 Mha of wheat and causing about 0.9 Mt loss of wheat production [8]. Severe epidemics of FHB also occurred in 1998 and 2003 on the North China Plain [9].

Over 17 species have been reported to be associated with FHB, with *Fusarium graminearum* species complex (FGSC) being the most common causal agent in many countries of the world including the USA, Canada, Australia, and in parts of central Europe [10,11], whereas in cooler, maritime regions such as the UK and northern Europe, *F. culmorum* predominates [12,13]. In Australia, *F. pseudograminearum* was also found to cause FHB in wheat and triticale in the northern wheatbelt of New South Wales in some years at low levels [4]. Recently, it has been confirmed that FGSC comprises at least 16 phylogenetically distinct species, with *F. graminearum* being the most widely distributed species causing FHB around the world [14,15], while *F. asiaticum* is the main FHB pathogen present in Asia [16,17,18,19]. From 1975 to 1985, the national FHB cooperation group of China surveyed the species associated with FHB in 22 provinces. More than 20 *Fusarium* species and subspecies were found to be associated with FHB but *F. graminearum* (FGSC) was the most important species in most provinces. However, *F. culmorum* was common in Guizhou province and the Ningxia Hui autonomous region [20]. In recent years, studies associated with FHB have shown that *F. graminearum* is the predominant species in the north of China where the annual average temperature is 15 °C or lower, whereas *F. asiaticum* is predominant in the south where the annual average temperature is above 15 °C and FHB epidemics occur most frequently [16,21,22,23]. 

Hebei province is located in the north of the North China Plain, an important agricultural region with biannual crop rotation of wheat and maize. In the last two decades, there have been major changes in farming systems in Hebei province. These changes include the adoption of continuous wheat–maize cropping systems with residue retention, and the introduction of high-yielding cultivars which lack resistance to FHB [9,24]. The major *Fusarium* species causing FHB, *F. graminearum*, also causes stalk and cob rot of maize, and produces abundant perithecia on wheat and maize residues. Presumably, these changes were conducive to an increase in the inoculum levels of *F. graminearum*, and other species that also cause FHB such as *F. culmorum* [1]. Most importantly, the adoption of wheat conservation farming systems accompanied by stubble retention, and drought during the growing season, favored the occurrence of Fusarium crown rot (FCR) of wheat caused by *F. pseudograminearum* in Hebei. Thus, it is not surprising that five isolates of *F. pseudograminearum* have also been detected from FHB samples [25]. 

Considering the major changes in the status of FHB and farming systems over the last decade, we aimed to assess possible changes in the pathogen composition of FHB, including the chemotypes of each species and their geographical distribution in Hebei province during the years 2005–2006 and 2013–2016. It is important to clarify the nature and proportion of the *Fusarium* species associated with FHB in Hebei province, as management decisions need to be made on the basis of such information. 

## 2. Results

### 2.1. Fusarium Species Determination

During the period 2005–2006, a total of 778 *Fusarium* isolates were recovered from rachis tissue affected by FHB collected from 42 sampling sites across 17 counties, and seven *Fusarium* species were identified after cultures were purified. *F. graminearum* was the most common species of the FHB complex recovered during the period 2005–2006. The percentage of isolates identified as *F. graminearum* ranged from 86.2% to 100% across the sampling sites, with an average of 96.7% (Table 1) and were recovered from all 42 sampling sites. *F. culmorum*, *F. equiseti*, *F. verticillioides*, *F. proliferatum*, *F. subglutinans* and *F. chlamydosporum* were also recovered from some of the samples with lower recoveries of 0.3%, 0.4%, 0.8%, 0.8%, 0.1% and 1.0%, respectively (Table 1). *F. graminearum* was the predominant pathogenic species recovered from all sampling sites during the period 2005–2006, and it was the only species isolated from samples from Hengshui and Tangshan regions. *F. culmorum* was only isolated from samples from Handan region with a recovery of 0.7%.

A total of 1002 purified *Fusarium* isolates were recovered from 122 sampling sites across 65 counties during the period 2013–2016. Five *Fusarium* species, *F. graminearum*, *F. asiaticum*, *F. pseudograminearum*, *F. culmorum* and *F. negundis*, were recovered and 978 out of 1002 (97.6%) isolates were identified as *F. graminearum*. The percentage of isolates identified as *F. graminearum* was 100% in Baoding and Langfang sampling sites (Table 2). *F. asiaticum*, *F. pseudograminearum*, *F. culmorum* and *F. negundis* were isolated at low recoveries of 0.7%, 1.4%, 0.2% and 0.1%, respectively (Table 2). Seven *F. asiaticum* isolates originated from Xingtai and Tangshan. Among them, six out of seven isolates were recovered from Tangshan, where wheat–maize and wheat–rice rotation systems coexisted. *F. pseudograminearum* was also recovered from most sampling sites except Baoding and Langfang. Isolates of *F. pseudograminearum* were mainly recovered from the mid-south of Hebei province, Handan, Xingtai, Hengshui, Cangzhou and Shijiazhuang where FCR had been found at high levels since 2013. Phylogenetic analysis of the TEF-1α gene sequence of *F. pseudograminearum* strains showed that there is no genetic difference between isolates from FHB and FCR (data not shown).

### 2.2. Prediction of Trichothecene Genotype

Seven hundred and fifty-two *Fusarium* isolates from 2005–2006 and 999 *Fusarium* isolates from 2013–2016 were chemotaxonomically classified into 15-ADON, 3-ADON and NIV types by primers 3CON/3NA/3D15A/3D3 or Tri13F/Tri13R, Tri303F/Tri303R, and Tri315F/Tri315R. All *F. graminearum* isolates from 2005–2006 were classified as 15-ADON type (Table 3). During the period 2013–2016, in the predominant species, *F. graminearum*, 15-ADON and 3-ADON types were both recovered. Among them, 971 out of 978 (99.3%) isolates were 15-ADON type, whereas only seven out of 978 (0.7%) isolates were classified as 3-ADON, which were isolated from the mid-south of Hebei province, Shijiazhuang, Xingtai and Handan. Moreover, all the isolates of *F. asiaticum* and of *F. pseudograminearum* were NIV and 3-ADON types, respectively.

## 3. Discussion

FHB can be caused by a number of fungal species. The most common pathogens are *F. graminearum*, *F. culmorum*, *F. avenaceum*, and *Microdochium nivale* [12]. Other fungi associated with this disease include *F. poae* [26], *F. pseudograminearum* (=*F. graminearum* Group 1 (Francis and Burgess)) [4] and *F. equiseti* [27]. *F. equiseti* is generally considered a saprophyte that can colonize diseased or senescent tissue. The relative contribution of these species to FHB will depend upon a range of variables, particularly temperature, rainfall and humidity [28]. In the present study, ten *Fusarium* species were confirmed to be associated with FHB in Hebei province. However, *F. graminearum* was found to be the dominant pathogenic species associated with FHB in the surveys over two consecutive periods of time, seven years apart. This finding is consistent with reports from North America and parts of Europe as well as other provinces of the northern part of China [1,23] where maize is now commonly used in rotation with wheat. Consequently, this rotation can be considered a key factor contributing to the high incidence of FHB caused by *F. graminearum* in Hebei. *F. graminearum* persists in maize and wheat residues and produces abundant perithecia on these residues. The perithecia release ascospores under cool wet conditions which act as the primary inoculum causing initial infection of the wheat heads [29]. Ascospores can also drift in the air and infect crops remote from the location of the perithecia [30,31]. 

In contrast, *F. culmorum* was recovered at a low level but was found during the periods 2005–2006 and 2013–2016. This finding also agrees with earlier findings which indicated that the species is more common in cooler conditions in areas where maize is less common such as cooler, maritime climate regions of Europe as well as Ningxia Hui autonomous region and Guizhou province of China [12,20]. In Hebei province, the wheat–maize rotation system enables the homothallic *F. graminearum* to produce abundant perithecia on maize stubble and then release ascospores as the primary inoculum of FHB. Climatic conditions also favor *F. graminearum* over *F. culmorum*, *F. avenaceum* and *F. poae*, as the former species has a higher optimum temperature, whereas the latter species prefer cooler conditions [28]. Several other *Fusarium* species were recovered during the period 2005–2006. In 2005–2006, the samples were collected slightly later after the onset of FHB. Presumably, this enabled more secondary colonizers to invade the heads prior to sampling and led to relatively higher isolation of a wider range of secondary colonizers. The isolation of *F. verticillioides*, *F. proliferatum* and *F. subglutinans* at low levels is not surprising as these are commonly associated with maize, as endophytes and stress-related stalk and cob rot pathogens. The remaining two *Fusarium* species, *F. equiseti* and *F. chlamydosporum,* are commonly associated with cereals as saprophytic species [32,33]. These species can interfere with the recovery of the pathogenic species following very wet conditions, even when a selective isolation medium is used. 

There were more pathogenic species detected during the period 2013–2016 than the period 2005–2006. *F. asiaticum* and *F. pseudograminearum* both emerged as causal agents of FHB in Hebei during the period 2013–2016. They were not recovered during the surveys in the period 2005–2006. *F. asiaticum* was isolated mainly from the regions with rice cultivation. This result is consistent with earlier findings from China, Korea and Brazil, in which *F. asiaticum* dominated in regions where rice is grown in rotation with wheat, whereas *F. graminearum* was more common in wheat–maize rotation systems [12,34,35,36]. Furthermore, *F. pseudograminearum* was also obtained from FHB-infected spikes in the mid-south part of Hebei province where FCR caused by *F. pseudograminearum* has emerged as a new destructive disease of wheat and the disease incidence and severity has been increasing year by year [37]. *F. pseudograminearum* has also been reported to cause FHB in Australia under favorable conditions, but usually at low levels [4,38]. Both *F. pseudograminearum* and *F. graminearum* caused FCR and FHB in surveys of wheat crops in Australia, though *F. pseudograminearum* was more frequently isolated from the crown, whereas *F. graminearum* was mostly recovered from the head [39]. In the wheat–maize biannual crop rotation system of Hebei province, 10–15 cm of standing wheat stubble is left on the field after harvest, then maize seed is sown directly. Subsequently, the maize and wheat stubbles are rotary-cultivated back into the upper 10 cm of soil after maize harvest. Wheat seeds are planted immediately, leaving no time for straw decomposition. This system provides more opportunity for the two pathogens, especially *F. pseudograminearum,* to infect the new host. Crop residues that favor persistence of *Fusarium* mycelia in infested residues also favor FCR [40], while the macroconidia of *F. pseudograminearum* produced on diseased wheat debris might also favor infection of wheat heads through water splash of macroconidia [41].

The composition of the pathogens causing FHB is believed to be determined, in part, by climatic factors such as temperature and moisture, as well as agronomic factors such as soil tillage and previous crops [42,43,44]. In the cooler, maritime climatic regions of Europe where *F. culmorum* has previously been reported to be the dominant pathogen of FHB, evidence shows that climatic and agricultural changes have favored the propagation of *F. graminearum* over *F. culmorum*, as the former species has a higher optimum temperature. Rising temperatures and an increase in minimum tillage and maize cropping have also favored *F. graminearum*. Similarly, there has been a decline in the presence of *F. culmorum* and an increase in *F. graminearum* in the UK [45], Netherlands [46], Denmark [47] and Poland [48] in the last two decades. Although the recovery of *F. pseudograminearum* from wheat heads has so far been identified in a limited amount in Hebei province, its potential development into a common pathogen of FHB is of concern in FCR and FHB co-existence regions. Thus, there is a need for ongoing monitoring of the incidence and distribution of *F. pseudograminearum* causing FHB in Hebei province. Studies need to span the entire FHB and FCR continuum to address key questions on pathogen biology and genetics in the local farming systems. Such information would be a valuable basis for clarifying the epidemiology and mycotoxin issues relating to FHB. It will also be important for the formulation of integrated disease management strategies for the management of the diseases caused by this fungus.

## 4. Materials and Methods 

### 4.1. Sampling

Diseased wheat spikes with characteristic symptoms of FHB were collected from farmers’ fields (42 sampling sites in 17 counties during the period 2005–2006 and 122 sampling sites in 65 counties during the period 2013–2016) in Hebei province. The sampling sites represented the different latitudes from the north to the south of Hebei and were recorded by a Global Positioning System (GPS) locator eTrex (Garmin Corporation, Xinbei City, Taiwan). The map indicating the sample sites was generated by the software ArcGIS10 (Esri, Redlands, CA, USA) (Figure 1). Each sample consisted of 10–20 spikes collected on an ad hoc basis across each field. 

### 4.2. Isolation and Morphological Identification

Discolored segments of floral rachis were surface sterilized with 70% ethanol and plated onto Peptone PCNB Agar (PPA), a selective medium for *Fusarium* species [49], and incubated at 22–25 °C under fluorescent and long-wavelength ultraviolet lights with a 12-h photoperiod [50]. All *Fusarium* isolates which developed from the rachis segments were sub-cultured to carnation leaf-piece agar (CLA) [51], and incubated as above for 10 d. *Fusarium* isolates were purified by single-spore isolation. Single-spore isolates were cultured on potato dextrose agar (PDA) and CLA. Isolates producing purple pigment and microconidia on PDA were also cultured on Spezieller Nahrstoffarmer Agar (SNA) [52]. Isolates were identified to species level on the basis of morphological criteria according to the descriptions in Burgess et al. [50] and Leslie & Summerell [52].

### 4.3. DNA Extraction

DNA of purified isolates was extracted using the commercial Fast-DNA^®^ Spin Kit (Qbiogene, Carlsbad, CA, USA). The mycelium of 5- to 7-day old cultures grown on PDA was scraped and added to Lysis Matrix A tubes and run in a FastPrep^®^ Cell Disrupter (Qbiogene, Carlsbad, CA, USA) for 20 s at speed level 5.5. The extraction was then carried out according to the manufacturer’s instructions.

### 4.4. Species Determination and Trichothecene Genotype Prediction

*Fusarium* species-specific PCR was performed using primers Fg16F/Fg16R, Fp1-1/Fp1-2 and Fc01F/Fc01R for the detection of FGSC, *F. pseudograminearum* and *F. culmorum* (Table 4). Fg16F/Fg16R produces a monomorphic product of 420 bp DNA fragment specific to *F. graminearum* and about 500 bp DNA fragment suspect to *F. asiaticum* [53,54]. Amplification reactions were carried out in 20 µL volumes containing 5–20 ng fungal DNA. The reaction mixture consisted of 0.2 mM concentrations of each deoxynucleoside triphosphate, 0.2 µm of each primer, and 1 unit Taq DNA polymerase in 1x PCR buffer with 1.5 mm MgCl_2_. A negative control, containing all reagents but no DNA, was used in every set of reactions. The cycling conditions consisted of denaturation (94 °C) for 30 s, annealing (Table 4) for 30 s and extension (72 °C) for 60 s followed by a final extension of 72 °C for 5 min. Isolates of other species and some of *F. graminearum* were amplified by primers EF-1α F/EF-1α R originated from partial translation elongation factor (*TEF-1α*) gene sequences (Table 4). The sequences of these strains were blasted in Fusarium-ID database [55] and the strains were confirmed to species level. 

For trichothecene genotype prediction, Tri3 multiplex amplification was performed in 20 µL volumes using primers 3CON/3NA/3D15A/3D3 for FGSC and Tri13F/Tri13R, Tri303F/Tri303R, and Tri315F/Tri315R for *F. pseudograminearum*, respectively (Table 4). The reaction mixture and the cycling conditions were the same as described above with the annealing temperatures listed in the table.

## Figures and Tables

**Figure 1 toxins-11-00243-f001:**
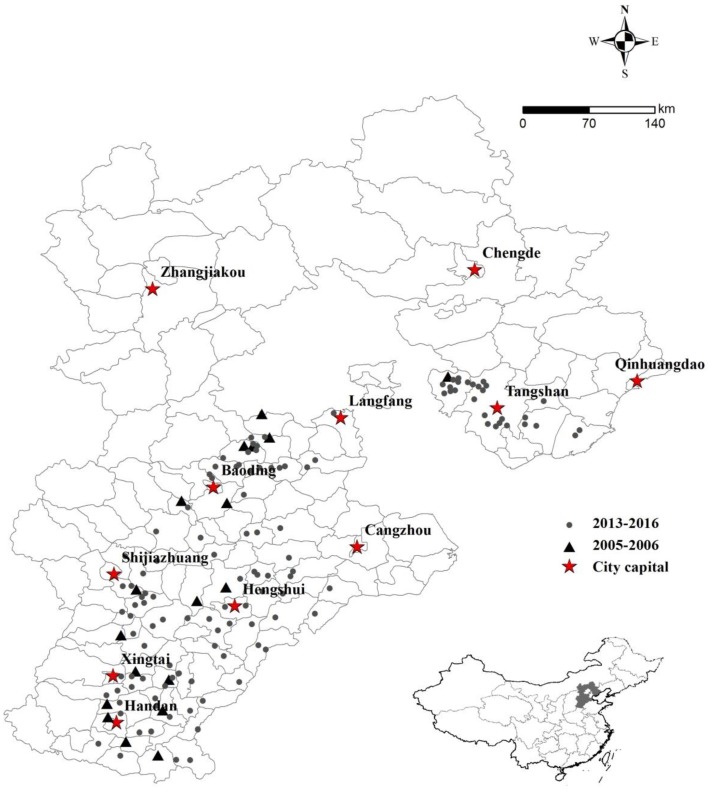
Map of Hebei province indicating the sampling sites during the periods 2005–2006 and 2013–2016.

**Table 1 toxins-11-00243-t001:** *Fusarium* species recovered from rachis tissue affected by Fusarium head blight in Hebei province during the period 2005–2006.

Sampling Sites	Isolates Obtained	Fg ^a^	Fc ^a^	Fe ^a^	Fv ^a^	Fp ^a^	Fs ^a^	Fch ^a^
Handan	280	274(97.9%) ^b^	2(0.7%)	2(0.7%)	0	0	0	2(0.7%)
Xingtai	87	75(86.2%)	0	0	5(5.7%)	3(3.4%)	0	4(4.6%)
Hengshui	15	15(100.0%)	0	0	0	0	0	0
Shijiazhuang	36	35(97.2%)	0	1(2.8%)	0	0	0	0
Baoding	349	342(98.0%)	0	0	1(0.3%)	3(0.9%)	1(0.3%)	2(0.6%)
Tangshan	11	11(100.0%)	0	0	0	0	0	0
Total	778	752(96.7%)	2(0.3%)	3(0.4%)	6(0.8%)	6(0.8%)	1(0.1%)	8(1.0%)

^a^ Fg: *F. graminearum*, Fc: *F. culmorum*, Fe: *F. equiseti*, Fv: *F. verticillioides*, Fp: *F. proliferatum*, Fs: *F. subglutinans*, Fch: *F. chlamydosporum*. ^b^ Number of isolates (isolation frequency).

**Table 2 toxins-11-00243-t002:** *Fusarium* species recovered from wheat rachis tissue affected by Fusarium head blight in Hebei province during the period 2013–2016.

Sampling Sites	Isolates Obtained	Fg ^a^	Fa ^a^	Fp ^a^	Fc ^a^	Fn ^a^
Handan	175	172(98.3%) ^b^	0	1(0.6%)	2(1.1%)	0
Xingtai	156	153(98.1%)	1(0.6%)	2(1.3%)	0	0
Hengshui	132	129(97.7%)	0	3(2.3%)	0	0
Cangzhou	18	16(88.9%)	0	2(11.1%)	0	0
Shijiazhuang	153	148(96.7%)	0	5(3.3%)	0	0
Baoding	157	157(100.0%)	0	0	0	0
Langfang	77	77(100.0%)	0	0	0	0
Tangshan	134	126(94.0%)	6(4.5%)	1(0.7%)	0	1(0.7%)
Total	1002	978(97.6%)	7(0.7%)	14(1.4%)	2(0.2%)	1(0.1%)

^a^ Fg: *F. graminearum*, Fa: *F. asiaticum*, Fp: *F. pseudograminearum*, Fc: *F. culmorum*, Fn: *F. negundis*. ^b^ Number of isolates (isolation frequency).

**Table 3 toxins-11-00243-t003:** Trichothecene genotype prediction of *Fusarium* isolates.

Sampling Sites	2005–2006		2013–2016				
	Fg ^a^	Total	Fg ^a^	Fg ^a^	Fa ^a^	Fp ^a^	Total
	15-ADON		15-ADON	3-ADON	NIV	3-ADON	
Handan	274	274	171	1	0	1	173
Xingtai	75	75	150	3	1	2	156
Hengshui	15	15	129	0	0	3	132
Cangzhou	-	-	16	0	0	2	18
Shijiazhuang	35	35	145	3	0	5	153
Baoding	342	342	157	0	0	0	157
Langfang	-	-	77	0	0	0	77
Tangshan	11	11	126	0	6	1	133
Total	752	752	971	7	7	14	999

^a^ Fg: *F. graminearum*, Fa: *F. asiaticum*, Fp: *F. pseudograminearum*.

**Table 4 toxins-11-00243-t004:** List of primers used for the species determination and trichothecene genotype prediction.

Primers	Sequence	Size (bp)	Annealing Temperature °C	Reference
Fg16F	CTCCGGATATGTTGCGTCAA	400–500	57	[56]
Fg16R	GGTAGGTATCCGACATGGCAA			
Fp1-1	CGGGGTAGTTTCACATTTCYG	523	57	[57]
Fp1-2	GAGAATGTGATGASGACAATA			
Fc01F	ATGGTGAACTCGTCGTGGC	570	62	[56]
Fc01R	CCCTTCTTACGCCAATCTCG			
EF-1α F	ATGGGTAAGGA(AG)GACAAGAC	700	56	[58]
EF-1α R	GGA(GA)GTACCAGT(GC)ATCATGTT			
3CON	TGGCAAAGACTGGTTCAC	840 or 610 or 243	52	[59]
3NA	GTGCACAGAATATACGAGC			
3D15A	ACTGACCCAAGCTGCCATC			
3D3A	CGCATTGGCTAACACATG			
Tri13F	TACGTGAAACATTGTTGGC	234 or 415	57	[60]
Tri13R	GGTGTCCCAGGATCTGCG			
Tri303F	GATGGCCGCAAGTGGA	583	52	[60]
Tri303R	GCCGGACTGCCCTATTG			
Tri315F	CTCGCTGAAGTTGGACGTAA	863	61	[60]
Tri315R	GTCTATGCTCTCAACGGACAAC			

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
