# Peer review of "Monitoring of Fusarium Species and Trichothecene Genotypes Associated with Fusarium Head Blight on Wheat in Hebei Province, China"

_toxins, 2019, doi:10.3390/toxins11050243_

Round 1

Reviewer 1 Report

This manuscript is an interesting report about monitoring the role of Fusarium species in FHB. The findings of this survey could give an integrated management of the diseases caused by these fungi.

I found this study appropriate for publication with some minor corrections and clarifications.

Why did you choose to investigate the specific periods in the first place?

Line 152: ‘’More Fusarium species were recovered during 2005-2006 than in 2013-2016’’. Why do you think of that?

Line 161: ‘’There were more pathogenic species detected during 2013-2016’’. Could you explain this finding?

Did you think of any extra statistical feature concerning your data?

Table 1 and 2: correct format of the words in the horizon axis. It is not clear of what the number in the parenthesis are representing. Pls clarify, either in the text or in the capture of the tables.

Table 3: is not mentioned in the text. Pls check

Author Response

Point 1: Why did you choose to investigate the specific periods in the first place?

Response 1: As Hebei province was a mild occurrence area of FHB in the early of 21st century. However, there was a severe epidemic of FHB in Hebei in year 2003. Therefore, it was the beginning of FHB epidemic in Hebei during the period of 2005-2006. In year 2012, Fusarium crown rot caused by Fusarium pseudograminearum was first found in wheat field of Hebei. Therefore, the period of 2013-2016 was chosen to investigate if F. pseudograminearum also infects the head of wheat in the field conditions.

Point 2: Line 152: “More Fusarium species were recovered during 2005-2006 than in 2013-2016”. Why do you think of that?.

Response 2: Because the samples were collected slightly later in 2005-2006. Presumably there was more secondary colonizers on the heads, which led to relatively higher isolation of a wider range of secondary colonizers. In order to express clearly, I have changed this sentence to “Several other Fusarium species were recovered during 2005-2006” in line 164-165 of the revised MS.

Point 3: Line 161: “There were more pathogenic species detected during 2013-2016”. Could you explain this finding?

Response 3: As wheat-maize rotation is the main cropping system in Hebei, F. graminearum was nearly the only pathogenic species detected during 2005-2006. Along with the occurrence of FCR and the rotation of wheat and rice, F. pseudograminearum and F. asiaticum have both detected as causal agents of FHB in Hebei during 2013-2016. Therefore, there were more pathogenic species detected during 2013-2016.

Point 4: Did you think of any extra statistical feature concerning your data?

Response 4: From the 2018 article by Yang: Host and Cropping System Shape the Fusarium Population: 3ADON-Producers Are Ubiquitous in Wheat Whereas NIV-Producers Are More Prevalent in Rice. Toxin 10:115. I personally presume there is no necessary for statistics of this kind of data.

Point 5: Table 1 and 2: correct format of the words in the horizon axis. It is not clear of what the number in the parenthesis are representing. Pls clarify, either in the text or in the capture of the tables.

Response 5: Table 1 and 2: Numbers are clarified in the notes of the tables.

Point 6: Table 3: is not mentioned in the text. Pls check.

Response 6: Table 3 is mentioned in line 130 of the text.

Reviewer 2 Report

The manuscript on the Fusarium species survey associated with FHB in Hebei is very well written, and the results and discussion of high quality. The only major concern the identified is the use of the word "chemotype". If toxin profiling has not been carried out, then the chemotype has not been proven. Genetic analyses suggests that the different species identified are of the described chemotypes, but without metabolic verification, these isolates can only be described as "genotypes". Please see the 2008 article by Desjardins: Natural product chemistry meets genetics: when is a genotype a chemotype? Journal of Agricultural and Food Chemistry 56:7587-7592. Toxin profiling would certainly enhance the manuscript, but is not necessary if the authors change the word chemotype to "genotype" or "trichothecene genotype" depending on the context. Additionally, it cannot be stated for example, these isolates are 3ADON producers" unless it has been shown using chemical analyses that the isolates do in fact produce 3ADON. Apart from this, only minor comments and suggestions are recommended prior to publication:

Tables: hard to read when brackets carry over. Should avoid words carrying over (ex. the s of isolates on second line) Can use Fg Fc etc. for species names. Or mayI be table should be landscape.

line 38 - why specifically winter cereals.

line 45 - "more recently" does not apply to the 80s and 90's, especially when there are many when the disease is still considered a major problem worldwide. The outbreaks in the 1980-1990s are of significance and should still be discussed, but should not be referred to as more recently.

line 77 - why "wheat crown rot (WCR)", should be "Fusarium crown rot (FCR) of wheat"; FCR is the common name for the disease, just as the wheat head blight is FHB of wheat.

line 171 -  Suggest removing the word recent since the citation is from 2004, and also, since FCR caused by F graminearum among other Fusarium spp. has been a problem in Australia for a number of years, even prior to 2004.

Line 191-193, the second clause of the sentence is a bit awkward "especially in WCR occurrence region is of concern"--I'm not positive on what is meant by this.

References: It is suggested that the authors go over their references and determine in some cases whether a more recent reference is suitable. For example, where the authors describe which Fusarium species predominate different parts of the world--it is likely that a more recent reference is available than the one from Parry et al 1995. This statement is still true, but it would be more convincing if it was also supported by a more recent survey.

Author Response

Point 1: About the concern of using the word "chemotype".

Response 1: I agree with the difference between chemotype and trichothecene genotype. However, despite the problems inherent in association studies, a number of linked TRI genes and intergenic regions have been proposed as markers for predicting trichothecene chemotypes and well used. In order to be more reasonable, I changed the chemotype determination to trichothecene genotype prediction in the text according to the 2011 article by Talas: Diversity in genetic structure and chemotype composition of Fusarium graminearum sensu stricto populations causing wheat head blight in individual fields in Germany. European Journal of Plant Pathology 131:39-48. Please find the details from the title, abstract, keywords, lines 125-137, 267, 280 and 282 of the revised MS.

Point 2: Tables: hard to read when brackets carry over. Should avoid words carrying over (ex. the s of isolates on second line) Can use Fg Fc etc. for species names. Or mayI be table should be landscape.

Response 2:   Table1 to table3 have been modified and table notes were added to avoid words carrying over. Please find the details in the revised MS.

Point 3: Line 38 - why specifically winter cereals.

Response 3:  Line 40 - winter cereals has been changed to wheat and other small grains.

Point 4: Line 45 - "more recently" does not apply to the 80s and 90's, especially when there are many when the disease is still considered a major problem worldwide. The outbreaks in the 1980-1990s are of significance and should still be discussed, but should not be referred to as more recently.

Response 4: Line 48 – “More recently” has been changed to “In the latest several decades”.

Point 5: line 77 - why "wheat crown rot (WCR)", should be "Fusarium crown rot (FCR) of wheat"; FCR is the common name for the disease, just as the wheat head blight is FHB of wheat.

Response 5: Wheat crown rot (WCR) in line 81 and others in the text have been revised to Fusarium crown rot (FCR).

Point 6: Line 171 - Suggest removing the word recent since the citation is from 2004, and also, since FCR caused by F graminearum among other Fusarium spp. has been a problem in Australia for a number of years, even prior to 2004.

Response 6:  Line 183 – Recently was removed.

Point 7: Line 191-193, the second clause of the sentence is a bit awkward "especially in WCR occurrence region is of concern"--I'm not positive on what is meant by this.

Response 7: Line 205-206, the sentence " its potential development into a common pathogen of FHB especially in WCR occurrence region is of concern" has been changed to “its potential development into a common pathogen of FHB is of concern especially in FCR and FHB co-existence regions”.   

Point 8: References.

Response 8:   Recent literatures has been added in line 58-59 and the references list. All the cited numbers have also been changed in the text and the references list.
